# Direct Base-Assisted C‒H Cyclonickelation of 6-Phenyl-2,2′-bipyridine

**DOI:** 10.3390/molecules25040997

**Published:** 2020-02-24

**Authors:** Nicolas Vogt, Vasily Sivchik, Aaron Sandleben, Gerald Hörner, Axel Klein

**Affiliations:** 1Department für Chemie, Institut für Anorganische Chemie, Universität zu Köln, Greinstraße 6, D-50939 Köln, Germany; 2Department of Chemistry, University of Eastern Finland, 80101 Joensuu, Finland; 3School of Chemistry, University of East Anglia, Earlham Road, Norwich NR4 7TJ, UK; 4Institut für Chemie, Anorganische Chemie IV, Universität Bayreuth, Universitätsstraße 30, D-95440 Bayreuth, Germany; gerald.hoerner@uni-bayreuth.de; 5Theoretische Chemie, Technische Universität Berlin, Straße des 17. Juni 135, D-10623 Berlin, Germany

**Keywords:** cyclometalation, cyclonickelation, C–H activation, organonickel, base-assisted

## Abstract

The organonickel complexes [Ni(Phbpy)X] (X = Br, OAc, CN) were obtained for the first time in a direct base-assisted arene C(sp^2^)–H cyclometalation reaction from the rather unreactive precursor materials NiX_2_ and HPhbpy (6-phenyl-2,2′-bipyridine) or from the versatile precursor [Ni(HPhbpy)Br_2_]_2_. Different from previously necessary C‒Br oxidative addition at Ni(0), an extended scan of reaction conditions allowed quantitative access to the title compound from Ni(II) on synthetically useful timescales through base-assisted C‒H activation in nonpolar media at elevated temperature. Optimisation of the reaction conditions (various bases, solvents, methods) identified 1:2 mixtures of acetate and carbonate as unrivalled synergetic base pairs in the optimum protocol that holds promise as a readily usable and easily tuneable access to a wide range of direct nickelation products. While for the base-assisted C‒H metalation of the noble metals Ru, Ir, Rh, or Pd, this acetate/carbonate method has been established for a few years, our study represents the leap into the world of the base metals of the 3d series.

## 1. Introduction

Metal-driven C(sp^2^)‒H activation and turnover of the resulting metalated species in catalytic processes has for a long time been the domain of heavy elements from the 4d and 5d rows, such as ruthenium, iridium and, especially, palladium. Platinum-promoted C‒H activation of hydrocarbons was one of the first milestones in this endeavour [1]. The intramolecular variant coined as cyclometalation initiated a period of rapid knowledge growth, owing to favourable metalation kinetics and thermodynamics [2,3]. Von Zelewsky et al. prepared the first biscycloplatinated derivatives of 2-phenylpyridine by transmetalation of the carbolithiated precursor back in the 1980s as well as discovered the intriguing luminescence of the obtained cyclometalated compounds [4]. Later, Constable et al. introduced cycloruthenation, cycloplatination, and cyclopalladation through C‒H activation (direct cyclometalation) of aryl/N-donor moieties [5,6]. This straightforward synthetic approach coupled with the potential diversity of π-conjugated derivatives of C^N ligands to tailor photophysical properties led to intensive development in the field with more than 1000 congeners published today [7,8,9,10,11,12,13,14,15,16,17,18,19]. For Pd(II) salts (e.g., chloride or acetate) the direct cyclometalation of 2-phenylpyridine proceeds already at room temperature [6,14,20]. Not surprisingly, a considerable number of π-conjugated C^N-derivatives are reported with interesting photophysical and biomedical properties [20,21]. Moreover, the ease of cyclopalladation resulted in the development of ground-breaking catalytic technologies based on Pd-catalysed directed C−H functionalisation [22,23,24,25].

In sharp contrast to the literally thousands of examples available for Pd(II) and Pt(II), cyclometalated Ni(II) derivatives of nitrogen-donating π-conjugated moieties have remained rare species, although the first example of a cyclometalated Ni(II) complex, [Ni(Cp)(*o*-phenylazo)-phenyl)], was prepared from [Ni(Cp)_2_] (Cp^−^ = cyclopentadienide) in 1963 [26]. Van Koten et al. introduced two approaches for the synthesis of Ni(N^C^N)-pincer complexes: (a) transmetalation of a Au(I) precursor; or (b) oxidative addition of zero-valent nickel to the bromoaryl precursor [27]. Much later, Wolczanski et al. directly metalated 2-phenylpyridine under harsh heating conditions (Scheme 1) [28], whereas Klein et al. performed a two-step cyclometalation via Pd-directed *ortho*-bromination of 6-phenyl-2,2′-bipyridine followed by the oxidative addition of zero-valent nickel, akin to van Koten’s approach [29,30]. Direct, base-assisted metalation with NiCl_2_ was demonstrated by Gong and Song in a pincer topology [31] and, in a similar approach by Zargarian et al. [32], it was found to be also feasible in macrocyclic N^C^N^C or N^C^N^N setups [33]. These reactions were considered to be electrophilic substitutions (H^+^ versus Ni(II)) that are archetypal for Pt(II) and Pd(II) complexes [1,25] but quite unique for Ni(II). Mechanistic insights were provided by van der Vlugt et al. for the formation of a [Ni(CNP)Br] pincer complex from the tetrahedral [Ni(HC^N^P)Br_2_] precursor using NEt_3_ as a base (Scheme 2) [34]. The isolation of a tri-coordinated T-shaped Ni complex with weak (an)agostic Ni^…^(H‒C) interaction invokes C‒H bond coordination to Ni(II), forming a sigma-complex as a key step of the process with base neutralising the proton outside of the coordination sphere.

Evidence for an inner-sphere attack of coordinated base was provided by Punji et al., who proposed the anion exchange of chloride or acetate by bis(trimethylsilyl)amide as a step precluding N-directed C‒H bond activation under solvent-free conditions [35] (Scheme 2). In keeping with this, Musaev et al. suggested a cesium/carboxylate cluster as the active basic principle based on computational studies [36]. The latter treatment appears to be the most comprehensive one to explain the “carboxylate effect”, which is a recurrent motif in metalation at d^8^ metal centres, as reviewed a while ago by Ackermann [37] and recently discussed in deep mechanistic detail by Macgregor et al. for Ru, Ir, Rh, or Pd systems [38,39,40]. Schafer, Love et al. elegantly adopted this motif for the C‒H activation of H_3_C‒N(Cy)C(O)N(H)(quinolin-8-yl), forming [Ni(‒CH_2_‒N(Cy)(O)N(quinolinyl)(PEt_3_)] [41].

Herein, we describe the development of a user-friendly protocol for a quantitative direct, base-assisted C(sp^2^)‒H cyclonickelation procedure, relying on a simple *ortho*-phenyl pyridine system and NiBr_2_ as the starting material. As a generic prototype of (di)nitrogen-directing moieties, we have selected 6-phenyl-2,2′-bipyridine (HPhbpy), rendering the protocol suitable for a wide range of bidentate and tridentate ligands. Bidentate directional group assisted C–H activations have recently gained high interest in organometallic catalysis [36,37,42,43,44,45]. In a thorough and comprehensive scan of the reaction conditions (solvent; counter ion; temperature; external base(s); Scheme 2 and Scheme 3) we identified the combination of high-boiling nonpolar solvents and acetate/carbonate as a homogeneous/heterogeneous base couple to enable rather rapid and very efficient nickelation.

## 2. Results

We worked out different procedures for the direct C‒H cyclonickelation of 6-phenyl-2,2′-bipyridine (HPhbpy) to yield the previously reported red complex [Ni(Phbpy)Br] (Scheme 3) [29,30].

The first synthesis attempts using NiBr_2_ and HPhbpy in boiling ethanol or methanol were unsuccessful. After the initial dissolution of the precursor materials, the typical red colour was observed. However, further reaction and isolation yielded grey-green materials. This is in line with earlier reports on the title complex [29,30] and related work on the complex [Ni(Mes)(bpy)Br] (Mes = 2,4,6-trimethylphenyl; bpy = 2,2′-bipyridine), which decomposes in protic solvents yielding greenish octahedral Ni(II) species [46]. Avoiding protic solvents and working in vigorously dried solvents, we heated HPhbpy with NiBr_2_ in toluene, but no yield was observed after 60 h at 130 °C (Table 1, entry 1). In keeping with the few literature reports, we ascribe the lack of reactivity to the lower basicity of the coordinated bromide ligand; similar observations have been reported by Campeau et al. [47] for a Pd catalysed bimolecular arylation and by Maseras and Echavarren et al. [48] for an intramolecular Pd-dependent arylation. In the following, we added external bases [49] such as NEt_3_, NaHCO_3_, Na_2_CO_3_, or Cs_2_CO_3_ to the reaction mixture (Entries 2–6). While for most reaction conditions the yields of [Ni(Phbpy)Br] were very poor (up to 2%), for Na_2_CO_3_/90 h/reflux, a crude yield of 19% was obtained (Table 1, Entry 6). In one of these experiments, MeOH was added to dissolve the starting materials, and the reaction time could be tremendously shortened (Entry 3). However, yield was poor due to decomposition, which was in line with the initial experiments. A further experiment in the melt (Entry 4) showed no yield.

During this series of experiments, we noticed the quantitative formation of the non-cyclometalated 1:1 adduct [Ni(HPhbpy)Br_2_]_2_ upon heating NiBr_2_ and HPhbpy in THF [50]. We were able to grow single crystals of this compound. Single-crystal XRD revealed a binuclear structure with an almost isotropic Ni_2_(µ-Br)_2_ diamond core with two triplet Ni(II) centres. Two other bromides are terminal ligands completing a five-coordinated trigonal bipyramidal geometry around nickel. (Figure 1; pertinent metrical data in Appendix A). In the following work, [Ni(HPhbpy)Br_2_]_2_ proved to be beneficial as a convenient, well-defined, and long-term storable form and was used as precursor (Table 1, Entries 7–9).

Starting from Ni(OAc)_2_^.^4H_2_O, the acetato complex [Ni(Phbpy)(OAc)] was obtained in traces (Entry 10), and the crystal and molecular structure could be solved (Figure 2, crystal and molecular data in Appendix A). The coordination geometry around Ni(II) is distorted square-planar.

In a further experiment, anhydrous nickel acetate was used in a solid-state reaction to produce the acetato complex, but the product could not be isolated due to the degradation by acetic acid (Entry 11). Quenching the same reaction mixture with cyanide yielded some crystals of [Ni(Phbpy)(CN)] (Entry 12, Appendix A). Unfortunately, the crystal quality was too low for a satisfying crystal structure solution and refinement. However, the atom connectivity is unequivocal. Thus, in contrast to Br^−^, acetate can obviously act as base.

Based on the encouraging 19% yield obtained from the reaction starting from the in situ formation of [Ni(HPhbpy)Br_2_]_2_ (Table 1, Entry 6) we started a series of reactions using isolated [Ni(HPhbpy)Br_2_]_2_. The reaction progress was monitored using UV-vis absorption spectroscopy with the spectrum of the previously reported [Ni(Phbpy)Br] [29] as reference. Using a 1:2 ratio and equimolar amounts (regarding the precursor) of KOAc and K_2_CO_3_, we achieved a yield of 43% within three days of heating in toluene (Table 2). The yields further increased to 91% upon prolonged heating in chlorobenzene, reaching 98% in 1,2-dichlorobenzene and quantitative yield in *p*-xylene within 25 h (Figure 3). The use of the high-boiling benzonitrile did not yield the product. After 20 h, no yield was observed and further heating at 200 °C for more than 60 h yielded a brown precipitated material that does not contain the targeted complex. We assume that the coordinating abilities of benzonitrile hamper the reaction by forming nitrile complexes [46], thus interfering with the formation of the intermediate [Ni(HPhbpy)Br_2_]_2_. This is in line with low yields for Ni-catalysed C(sp^2^)−H and C(sp^3^)−H functionalisation of aminoquinoline in polar solvents such as DMSO, dioxane, and MeCN [51].

A detailed look on the time-dependence of the reaction for selected examples (Figure 3) shows that *p*-xylene and 1,2-dichlorobenzene reach high yields already after about 20 h, while chlorobenzene requires at least 40 h. Using NiBr_2_ instead of [Ni(HPhbpy)Br_2_]_2_ for the reaction in *p*-xylene under the same conditions afforded only 18% yield after 20 h but could be brought to 93% yield when heating for 64 h. This observation is in line with the assumed pre-formation of the *p*-xylene-soluble [Ni(HPhbpy)Br_2_]_2_ from insoluble NiBr_2_. Finally, a blank experiment in 1,2-dichlorobenzene (no external base) delivered a mediocre 15% yield after 64 h.

## 3. Discussion

Summarising the experimental results shows that (*i*) protic solvents and water-containing Ni salts must be avoided. Although they speed up reaction times due to perfect dissolution of the precursor compounds especially simple Ni(II) salts, the final product is quenched, presumably forming non-cyclometalated hexacoordinate Ni(II) species. (*ii*) The acetate/carbonate mixture is capable of driving the C‒H metalation to completion, whereas single-component bases such as CO_3_^2−^, HCO_3_^−^, ^−^O*t*Bu, and NEt_3_ promote the reaction on a trace level. (*iii*) The reaction proceeds preferably via the formation of the N^N-coordinated intermediate [Ni(HPhbpy)Br_2_]_2_. (*iv*) This precursor allows the use of high boiling nonpolar solvents, while protic solvents and water-containing polar solvents, which are necessary to dissolve the NiBr_2_ precursor, can be avoided.

The observation of high reaction temperatures suggests a high activation barrier for the direct cyclonickelation of N-donor/aryl systems. Similarly, cyclonickelation to form C^N or N^C^N chelates required harsh heating [28,31,32]. High reaction temperatures seem to be the price for starting from rather unreactive Ni(II) precursors and a not very reactive C(sp^2^)–H bond. In addition, the base-assisted N–H_2_C(sp^3^)–H nickelation yielding [Ni(‒CH_2_‒N(Cy)(O)N(quinolinyl)(PEt_3_)] required temperatures above 70° [41]. In contrast to this, the C^N^P nickelation in the recently reported complex [Ni(HPh‒Py‒O‒PR_2_)Br_2_] (R = tBu) proceeds at 50 °C in an hour [34]. The significantly lower activation barrier can be tentatively explained by the following factors: (*i*) the P-donor brings the Ni(II) centre closer to the C–H activation site to enable (an)agnostic interactions, since phosphorus has a bigger covalent radius versus nitrogen (107 pm versus 71 pm), resulting in a Ni–P bond length of approximately 2.29 Å compared with the Ni–N bond of approximately 2.04 Å; (*ii*) this P-donor (decorated by two *t*Bu groups) is a much stronger σ-donor and perhaps a better π-acceptor than the N-site in our study. The Ni^…^C distance in our precursor is 2.97(1) Å, which is quite short compared to the 3.039(3) Å of the phosphine complex. However, this distance is largely depending on the angle of the freely rotating phenyl group toward the binding plane, and the values represent only the solid state and not the molecules in solution. Moreover, the Ni–C bond in the cyclonickelated species [Ni(Ph‒Py‒O‒PR_2_)Br] [34], [Ni(Phbpy)Br] [29], and [Ni(Phbpy)OAc] (Appendix A) is of the same magnitude (approximately 1.9 Å). Therefore, the crystallographic data suggest that the difference in the electronic properties of the P- and N-donors rather than steric effects are responsible for the much lower activation barrier.

It is tempting to correlate the enhanced reaction efficiency with the high boiling points of these solvents, indicating simple Arrhenius behaviour. However, the complete lack of reactivity in benzonitrile as a solvent with a similarly high boiling point indicates that aspects of solvent polarity and coordination ability also affect the reaction progress. This generally agrees with the recent work of Sandford et al., who reported low cyclonickelation yields in polar-coordinating solvents [51]. In our study, alongside the series of solvents, an increase in polarity ($E_{T}^{N}\;\mathrm{values})$ [52] leads to a decrease in activity. This inverse relationship points to a nonpolar transition state of the cyclometalation reaction, in stark contrast to the findings of Davies and Macgregor et al. [39], who inferred from substituent Hammett plots a substantial accumulation of positive charge in the transition state of a cycloruthenation reaction. Thus, in future work, we will model this reaction using density functional theory (DFT) methods.

## 4. Materials and Methods

### 4.1. Materials

Commercially available chemicals were purchased from Sigma-Aldrich, Acros, ABCR, or Fisher-Scientific and were used without further purification. Dry THF was obtained from distillation over sodium/potassium alloy. The preparation of *N*-(1-(2-pyridyl)-1-oxo-2-ethyl)pyridinium-iodide (Kroehnke salt) and 6-Phenyl-2,2′-bipyridine (HPhbpy) is described in the Appendix A.

**Preparation of [Ni(HPhbpy)Br_2_]_2_** (adopted from Yang et al.) [50]: NiBr_2_ (328 mg, 1.50 mmol), HPhbpy (348 mg, 1.50 mmol) in THF (125 mL) were heated under reflux for 15 h. After cooling to room temperature, a yellow solid was collected by filtration on a glass frit, washed using diethyl ether (3 × 10 mL), and dried in vacuo for 15 h. Yield: 72% (487 mg). Elemental analysis found (calc. for C_16_H_12_Br_2_N_2_Ni) C 42.62 (42.63); H 2.86 (2.68); N 6.23 (6.21)%.

### 4.2. Cyclonickelation Experiments

**General considerations:** In comparison with Pt or Pd congeners, cyclometalated Ni(II) complexes are less stable due to weaker Ni‒C and Ni‒N bonds [38,53]. The Ni–N bond, being also weaker than Ni–P bonds, is susceptible to facile ligand substitution by oxygen donors (e.g., alcohols) even in the case of chelating N^N-donors. [46]. The Ni–C bond is also chemically unstable. In addition to facile hydrolysis [46], oxygen insertion can occur in the presence of dioxygen with the formation of C‒O‒Ni bonds [54,55]. Thus, the cyclonickelation in this study was conducted under N_2_ atmosphere in solvents purified by distillation over Na-benzophenone ketyl (toluene or THF) or rigorously dried molecular sieves (chlorobenzene, 1,2-dichlorobenzene, and benzonitrile).

**Preparation of [Ni(Phbpy)Br] via C‒H-activation:** After a number of optimisation experiments (Table 1 and Table 2), the reaction was carried out as follows. Under inert atmosphere [Ni(HPhbpy)Br_2_]_2_ (112.7 mg, 0.25 mmol, 1 eq.) was suspended in *p*-xylene (140 mL). After adding KOAc (25.5 mg, 0.25 mmol, 1 eq.) and K_2_CO_3_ (34.6 mg, 0.25 mmol, 1 eq.), the mixture was heated to reflux. For the separation of forming water, the reaction flask was interconnected by a glass frit filled with activated molecular sieve to the reflux condenser. After 25 h, the mixture was allowed to cool to room temperature, and the solvent was evaporated under reduced pressure. The deep red residue was dissolved in dry CH_2_Cl_2_ (100 mL) and filtered through a plug of Na_2_SO_4_/Celite. After evaporating the solvent, the product was received as a red powder. The yield was determined from the UV-vis absorption band of the product [Ni(Phbpy)Br] in THF (see Appendix A). The isolated material (183 mg, 0.495 mmol, 99%) was identified through its ^1^H NMR spectrum [29,30].

From previous work, it was known that the [Ni(Phbpy)Br] is somewhat stable and soluble in CH_2_Cl_2_ (^1^H NMR) and THF (reaction medium, crystallisation solvent). The complex has a characteristic red colour [29,30]. In the course of our investigation, we found that [Ni(Phbpy)Br] is insoluble in diethyl ether. Profiting from its moderate solubility and stability in acetone, ^1^H and ^1^H-^1^H COSY NMR spectra was measured in acetone-d^6^. Additionally, the compound can be recrystallised by the vapour diffusion of diethyl ether to CH_2_Cl_2_ or acetone at room temperature without special precautions. At the final stage, the red crystals can be quickly washed using methanol to eliminate the contaminant (yellow powder, presumably NiBr_2_). Yet, [Ni(Phbpy)Br] degrades in methanol solution, as can be seen from the colour change from red to yellow and green, and ^1^H NMR (CD_3_OD) gave very broad signals indicating the formation of paramagnetic species. Details on the cyclometalation reactions yielding [Ni(Phbpy)X] (X = Br, OAc, CN) through base-assisted C‒H activation (direct cyclometalation) are available in the Appendix A.

**Instrumentation:**^1^H, ^13^C and ^19^F NMR spectra were recorded on a Bruker Avance II 300 MHz (^1^H: 300 MHz, ^13^C: 75 MHz, ^19^F: 282 MHz, Ettlingen, Germany) equipped with a double resonance (BBFO) 5 mm observe probehead with z-gradient coil or on Bruker Avance 400 spectrometer (^1^H: 400 MHz, ^13^C: 100 MHz, Ettlingen, Germany). Chemical shifts were relative to TMS. UV–vis absorption spectra were recorded on a Varian Cary 05E spectrophotometer (Troisdorf, Germany). Elemental analyses were obtained using a HEKAtech CHNS EuroEA 3000 analyzer (Wegberg, Germany). EI-MS spectra were measured using a Finnigan MAT 95 mass spectrometer (Bremen, Germany). Simulations were performed using ISOPRO 3.0 (Sunnyvale, CA, USA). Single crystal structure analysis (XRD): Measurement of [Ni(HPhbpy)Br_2_]_2_ was performed at 170(2) K using an IPDS IIT (STOE and Cie., Darmstadt, Germany) diffractometer, with Mo-Kα radiation (λ = 0.71073 Å) employing ω-φ-2θ scan technique. The structure was solved by direct methods using SIR 2014 [56], and refinement was carried out with SHELXL 2018 employing full-matrix least-squares methods on *F*_0_^2^ ≥ 2σ(*F*_0_^2^) [57]. The numerical absorption corrections (X-RED V1.22; Stoe & Cie, 2001, Darmstadt, Germany) were performed after optimising the crystal shapes using X-SHAPE V1.06 (Stoe & Cie, 1999, Darmstadt, Germany) [58]. The non-hydrogen atoms were refined with anisotropic displacement parameters without any constraints. The hydrogen atoms were included by using appropriate riding models. Measurement of [Ni(Phbpy)(OAc)] was conducted at 150 K. The X-ray diffraction data were collected with Bruker Kappa Apex-II (Ettlingen, Germany) diffractometer by using Mo-Kα radiation (λ = 0.71073 Å). The APEX2 [59] program package was used for cell refinements and data reduction. Using OLEX2 [60], the structure was solved with the SHELXS program structure solution program using direct methods and refined with the SHELXL refinement package using least squares minimisation [61]. A numerical absorption correction (SADABS) [62] was applied to all data. The non-hydrogen atoms were refined with anisotropic displacement parameters without any constraints. All H atoms were positioned geometrically and constrained to ride on their parent atoms, with C−H = 0.95−0.99 Å and U_iso_ = 1.2−1.5 U_eq_ (parent atom). Data of both structure solutions and refinements can be obtained free of charge at https://summary.ccdc.cam.ac.uk/structure-summary-form or from the Cambridge Crystallographic Data Centre, 12 Union Road, Cambridge, CB2 1EZ, UK (fax: +44-1223-336-033 or e-mail: deposit@ccdc.cam.ac.uk). CCDC 1956740 ([Ni(HPhbpy)Br_2_]_2_) and 1956739 ([Ni(Phbpy)(OAc)]).

## 5. Conclusions

Herein, we have addressed the central problem of a direct metalation of arene C(sp^2^)‒H functions using rather unreactive Ni(II) precursors such as NiBr_2_ or Ni(OAc)_2_, which lifts the limitation of user-friendly metalation routines to the 4d and 5d rows. The target organonickel complex [Ni(Phbpy)Br] (HPhbpy = 6-phenyl-2,2′-bipyridine), which previously was accessible only from C‒Br activation through a Ni(0) precursor, has now been obtained for the first time through C‒H activation in a direct, base-assisted cyclonickelation. Optimal reaction conditions were identified to require (i) anhydrous precursor materials, ii) elevated temperature, (iii) nonpolar media (*p*-xylene, (di)chlorobenzene), and (iv) the presence of carbonate and acetate as a synergetic base pair. On the other hand, the di-nitrogen coordination step of NiX_2_ (X = Br, OAc) proceeds slowly in nonpolar media. Therefore, [Ni(HPhbpy)Br_2_]_2_ obtained by Ni(II) coordination in THF proved to be a convenient precursor for the direct, base-assisted metallation. The concerted fulfilment of all four criteria allows quantitative direct C‒H activation on a synthetically reasonable timescale.

Although the reaction requires high temperatures > 100 °C, overall, the described quantitative direct nickelation appears superior to previous protocols with respect to a potential diversification. Variation of the substitution pattern of the metalated arene including deuteration and DFT modelling is subject of ongoing work. The high efficiency motivates a future extension to the more challenging cyclometalation of C(sp^3^)‒H protoligands, which was elegantly demonstrated recently by Schafer, Love et al. for the formation of [Ni(‒CH_2_‒N(Cy)C(O)N(quinoline-8-yl)(PEt_3_)] [41]. While for the noble metals Ru, Ir, Rh, or Pd, the base-assisted C(sp^2^)‒H deprotonation and metalation using acetate/carbonate has been established for a few years [38,39,40] our study represents an important step into the world of the base metals of the 3d series.

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
