# Peer review of "Direct Base-Assisted C‒H Cyclonickelation of 6-Phenyl-2,2′-bipyridine"

_molecules, 2020, doi:10.3390/molecules25040997_

Round 1
Reviewer 1 Report
This paper by Vogt et al. presents well-documented research on organometallic chemistry of Ni(II). Activation of C-H bonds with organometallic species is of major importance in catalysis, the main advantage of such process should be mild conditions and selectivity, with oxidative addition and σ-bond metathesis as the most important mechanistic pathways. In this work, harsh conditions, extended reaction times and a base (stochiometric amount) are used, thus the key advantages are not attained. Therefore, in my opinion, the title of this manuscript is misleading, and “base-assisted” must be added. Similarly, the amount of the base used in each run should be clearly provided in Table 1.
Other minor corrections:
Caption to Figure 1: C(sp2)-H: superscript is missing
Page 2, line 56: full stop is missing before “Van Koten”
Page 4, line 113: “In the following …”, unclear, please correct this sentence
Table 1, run 1: bp of toluene is 110 ºC while temperature 130 ºC is reported; this is confusing, if indeed “Solvent evaporated in reflux”, as given in the SI, this run is doubtful and should be removed or repeated in a way that the solvent does not escape.
Table 1: “visual” analysis is a very poor scientific method, there are many red complexes, and NMR of the recovered material should be measured, in all cases with yield is 0 (what do you mean by “-“ ?) the amount of recoverd substrate should be reported
Page 4, line 136: “storage form”, unclear, what do you mean?
Page 5, line 151: “pre-dried”: what do you precisely mean? Is it anhydrous?
Figure 3: Generally “conversion” refers to a substrate, not to a reaction, so conversion of “name of substrate” should be reporetd, in the case of UV-vis spectroscopy, I suppose, a calibration curve is necessary to give guantitative results (I could not find a calibration curve). Thus, the caption would be more appropriate as "Concentration of "name of product" in cyclometallation reaction ..."
Author Response
Reviewer 1
This paper by Vogt et al. presents well-documented research on organometallic chemistry of Ni(II). Activation of C-H bonds with organometallic species is of major importance in catalysis, the main advantage of such process should be mild conditions and selectivity, with oxidative addition and σ-bond metathesis as the most important mechanistic pathways. In this work, harsh conditions, extended reaction times and a base (stochiometric amount) are used, thus the key advantages are not attained. Therefore, in my opinion, the title of this manuscript is misleading, and “base-assisted” must be added. Similarly, the amount of the base used in each run should be clearly provided in Table 1.
Response 1: The reviewer is right, that our reaction conditions are not mild. However, any C-H activation using Ni requires either very reactive Ni species or very reactive C-H group, which has been demonstrated in some cases before. We present a case for a quite unreactive arene and a very unreactive Ni(II) source. The metalation is only supported through the directing groups but needs high thermal activation.
So, it seems that when doing C-H metalation with Ni(II) one price has to be paid. We have added some comments to the abstract, Discussion, and Conclusion sections accordingly.
Response 2: We changed the title to: Direct Base-Assisted C‒H Cyclonickelation of 6-Phenyl-2,2’-bipyridine.
Response 3: The exact amount of base was added to Table 1.
Other minor corrections:
Caption to Figure 1: C(sp2)-H: superscript is missing
Response: corrected
Page 2, line 56: full stop is missing before “Van Koten”
Response: corrected
Page 4, line 113: “In the following …”, unclear, please correct this sentence
Response: the sentence was re-written
Table 1, run 1: bp of toluene is 110 ºC while temperature 130 ºC is reported; this is confusing, if indeed “Solvent evaporated in reflux”, as given in the SI, this run is doubtful and should be removed or repeated in a way that the solvent does not escape.
Response: We have changed Table 1. In the previous version we gave the bath temperatures not the temperature of the reaction mixture, which explains the discrepancy. It’s true that this was confusing. We have now entered the reaction mixture T (e.g. 111°C for refluxing toluene).
Table 1: “visual” analysis is a very poor scientific method, there are many red complexes, and NMR of the recovered material should be measured, in all cases with yield is 0 (what do you mean by “-“ ?) the amount of recoverd substrate should be reported
Response: we agree with the reviewer. We have checked the records again and modified Table 1 and Table S1. In all cases where no yield was obtained, the visual analysis is probably reliable, the red colour of the formed complex cannot be overseen. The “-“ was replaced by 0. Unfortunately, the residues were not studied in these cases.
Page 4, line 136: “storage form”, unclear, what do you mean?
Response: “Storage” has been replaced by “long-term stable storable”
Page 5, line 151: “pre-dried”: what do you precisely mean? Is it anhydrous?
Response: Yes, we have changed this.
Figure 3: Generally “conversion” refers to a substrate, not to a reaction, so conversion of “name of substrate” should be reporetd, in the case of UV-vis spectroscopy, I suppose, a calibration curve is necessary to give guantitative results (I could not find a calibration curve). Thus, the caption would be more appropriate as "Concentration of "name of product" in cyclometallation reaction ..."
Response: The reviewer is right. We have changed “conversion” to “yield” and have added the following sentence to Table 2 to make clear that we based our yields on the long-wavelength absorption of the target complex. Thus, no calibration curve was necessary:
“Yields as observed from UV-vis absorption measurements of the reaction solutions were based on the reported molar extinction coefficients of the long-wavelength maximum around 510 nm of the target complex [Ni(Phbpy)Br] [29,30]”
Reviewer 2 Report
Sivichik and Klein and coworkers reported that C–H nickelation of 6-phenyl-2,2'-bipyridine. Although several reports on C–H nickelation of arenes with nitrogen-based directing groups existed, actually the methods using 6-phenyl-2,2'-bipyridine is unknown. Previously, Klein and coworkers have reported the synthesis of same nickel complex through C-Br oxidative additions to nickel. The present method in this manuscript would be superior to the previous method, avoiding pre-functionalization of Ph-bipyridine core. In this work, although the result would seem relatively predictable in my eye, the group realized the direct synthesis of this nickel complex using C-H nickelation by finding out the conditions with KOAc/K2CO3 in p-xylene. Because the method still requires pre-complexation step before C-H nickelation (actually two-step procedure), this method can be considered as indirect method for the synthesis of this, in my eye. This reviewer can support the publication of the work if “direct” synthesis of the complex just in one-step starting from 6-phenyl-2,2'-bipyridine.
1) Discussion around Table 1 should be fully revised. This is really confusing for readers. In my opinion, entries 1–6 can be moved to SI. It might be possible to discuss starting from the findings of the formation of [Ni(HPhbpy)Br2]2.
2) Page 5, line 164, “Surprisingly”. What is surprising result? Benzonitrile and xylenes are completely different solvent in terms of coordinating ability to metals. This sentence should be revised.
3) It is curious that the reaction of NiBr2 with 6-phenyl-2,2'-bipyridine in the presence of K2CO3/KOAc in p-xylene under reflux conditions.
4) It would be better to discuss the mechanism of C-H nickelation step. Is this CMD type or sigma-bond metathesis type? The role of base is completely unclear at this stage. Hammett plots and KIE experiments would be helpful to discuss this.
Author Response
Reviewer 2
Sivichik and Klein and coworkers reported that C–H nickelation of 6-phenyl-2,2'-bipyridine. Although several reports on C–H nickelation of arenes with nitrogen-based directing groups existed, actually the methods using 6-phenyl-2,2'-bipyridine is unknown. Previously, Klein and coworkers have reported the synthesis of same nickel complex through C-Br oxidative additions to nickel. The present method in this manuscript would be superior to the previous method, avoiding pre-functionalization of Ph-bipyridine core. In this work, although the result would seem relatively predictable in my eye, the group realized the direct synthesis of this nickel complex using C-H nickelation by finding out the conditions with KOAc/K2CO3 in p-xylene. Because the method still requires pre-complexation step before C-H nickelation (actually two-step procedure), this method can be considered as indirect method for the synthesis of this, in my eye. This reviewer can support the publication of the work if “direct” synthesis of the complex just in one-step starting from 6-phenyl-2,2'-bipyridine.
Response: We have changed the title to: Direct Base-Assisted C‒H Cyclonickelation of 6-Phenyl-2,2’-bipyridine
1) Discussion around Table 1 should be fully revised. This is really confusing for readers. In my opinion, entries 1–6 can be moved to SI. It might be possible to discuss starting from the findings of the formation of [Ni(HPhbpy)Br2]2.
Response: The discussion about the experiments entries 1 to 7 has been revised, but we want to keep it in the manuscript because it demonstrates the utility of the [Ni(HPhbpy)Br2]2 precursor outlined later in the Discussion section. Moreover, this part is only about 10 lines.
2) Page 5, line 164, “Surprisingly”. What is surprising result? Benzonitrile and xylenes are completely different solvent in terms of coordinating ability to metals. This sentence should be revised.
Response: The sentence was revised: The use of the high-boiling benzonitrile …
And we have added as explanation: We assume that the coordinating abilities of benzonitrile hamper the reaction by forming nitrile complexes [46], thus interfering with the formation of the intermediate [Ni(HPhbpy)Br2]2.
In line 222 we also added “solvent polarity and coordination ability”
3) It is curious that the reaction of NiBr2 with 6-phenyl-2,2'-bipyridine in the presence of K2CO3/KOAc in p-xylene under reflux conditions.
Response: This is a very good idea and we expected this question. Thus, we have carried out this experiment and report it now: Using NiBr2 instead of [Ni(HPhbpy)Br2]2 for the reaction in p-xylene under the same conditions afforded only 18% yield after 20 h but could be brought to 93% yield when heating for 64 h. This observation is in line with the assumed pre-formation of the p-xylene-soluble [Ni(HPhbpy)Br2]2 from insoluble NiBr2.
4) It would be better to discuss the mechanism of C-H nickelation step. Is this CMD type or sigma-bond metathesis type? The role of base is completely unclear at this stage. Hammett plots and KIE experiments would be helpful to discuss this.
Response: As announced in the Conclusions section we will study the mechanism in detail. We have also outlined there: “…in stark contrast to the findings of Davies and Macgregor et al. [39] who inferred from substituent Hammett plots a substantial accumulation of positive charge in the transition state of a cycloruthenation reaction.” It is presumably AMLA/CMD type but we do not want to speculate now.
Reviewer 3 Report
Molecules
Manuscript ID: molecules-724802
Direct C‒H Cyclonickelation of 6-Phenyl-2,2’-bipyridine
Axel Klein * , Nicolas Vogt , Vasily Sivchik * , Aaron Sandleben , Gerald Hoerner
This manuscript deals with a new and efficient synthesis of [Ni(Phbpy)X] with X = Br, OAc and CN via C-H cyclonickelation from [Ni(HPhbpy)Br2]2. The obtained results are interesting and well discussed and a detailed study on the optimization of the experimental conditions is reported. Hence I support the acceptance of this work with minor revisions:
1) some typos: line 43 ,44,56: dot is missing
2) In Table 2 ,(entry 13), the reaction is reported to be performed in toluene for 72h. In line 161, when the same reaction is described in the text, it is written that the reaction has been carried out for 2 days. Please check which is the right duration.
Author Response
Reviewer 3.
This manuscript deals with a new and efficient synthesis of [Ni(Phbpy)X] with X = Br, OAc and CN via C-H cyclonickelation from [Ni(HPhbpy)Br2]2. The obtained results are interesting and well discussed and a detailed study on the optimization of the experimental conditions is reported. Hence I support the acceptance of this work with minor revisions:
1) some typos: line 43 ,44,56: dot is missing
Response: corrected
2) In Table 2 ,(entry 13), the reaction is reported to be performed in toluene for 72h. In line 161, when the same reaction is described in the text, it is written that the reaction has been carried out for 2 days. Please check which is the right duration.
Response: corrected, its three days = 72 h
Round 2
Reviewer 2 Report
Revised manuscript by Sivichik and Klein was actually somewhat improved from the original report. Although it is required long reaction time, the success of C-H nickelation starting from NiBr2 in the presence of K2CO3/KOAc in p-xylene is noteworthy. With this status of this manuscript, this reviewer can support the publication of this work in Molecules.
This reviewer is highly looking forward to C-H nickelation mechanism being unveiled in a next report somewhere.